

# A public data set of human balance evaluations

Damiana A. Santos[*] and Marcos Duarte[*]

Biomedical Engineering, Federal University of ABC, São Bernardo do Campo, SP, Brazil
[*] These authors contributed equally to this work.

## ABSTRACT

The goal of this study was to create a public data set with results of qualitative and quantitative evaluations related to human balance. Subject's balance was evaluated by posturography using a force platform and by the Mini Balance Evaluation Systems Tests. In the posturography test, we evaluated subjects standing still for 60 s in four different conditions where vision and the standing surface were manipulated: on a rigid surface with eyes open; on a rigid surface with eyes closed; on an unstable surface with eyes open; on an unstable surface with eyes closed. Each condition was performed three times and the order of the conditions was randomized. In addition, the following tests were employed in order to better characterize each subject: Short Falls Efficacy Scale International; International Physical Activity Questionnaire Short Version; and Trail Making Test. The subjects were also interviewed to collect information about their socio-cultural, demographic, and health characteristics. The data set comprises signals from the force platform (raw data for the force, moments of forces, and centers of pressure) of 163 subjects plus one file with information about the subjects and balance conditions and the results of the other evaluations. All the data is available at PhysioNet and at Figshare.

Corresponding author
Marcos Duarte, duartexyz@gmail.com

## INTRODUCTION

Age-related disabilities and certain illnesses affect balance in humans and can negatively influence their health and quality of life. There has been great effort by researchers and clinicians for a greater understanding of this problem and postural control in humans has been under intense scientific investigation over the past decades.

There is a large variety of tests to describe balance in humans (for a review see *Duarte & Freitas, 2010*; *Paillard & Noe, 2015*; *Scoppa et al., 2013*; *Shumway-Cook & Woollacott, 2001*; *Visser et al., 2008*; *Yamamoto et al., 2015*). The most common quantitative measurement to characterize body balance is the displacement of the center of pressure (COP); the point of application of the resultant vertical forces acting on the subject's surface of support. COP displacement is typically measured with a force plate and presented as time-series of numerical data in the anterior-posterior (ap) and medio-lateral (ml) directions in relation to the subject's orientation. The technique concerned with the measurement of COP displacement in this context is commonly referred to as stabilography or posturography.

In stabilography, there is no consensus yet on the best techniques to analyze COP displacement in order to extract meaningful information about the subject's balance. There are also numerous different protocols (which include the type of task the subject performs, instructions to the subject, duration of the task, type of instrument, etc.) for data collection, algorithms, and variables to process and characterize COP displacement (for a review on these issues see *Duarte & Freitas, 2010*; *Paillard & Noe, 2015*; *Ruhe, Fejer & Walker, 2010*; *Scoppa et al., 2013*; *Visser et al., 2008*; *Yamamoto et al., 2015*). Part of the lack of consensus and conflicting findings might be because of the typically large intra and inter-subject variability of the COP displacement data, and that some of the studies present low statistical power (typically because of a small sample size) (*Ruhe, Fejer & Walker, 2010*; *Samson & Crowe, 1996*). A related problem, not specific to the field of stabilography, is that researchers propose and compare new methods of analysis based on data from different subjects across centers.

The deployment of a public data set of human balance evaluations would allow the access of a normative reference for data comparison and testing analysis from different centers. In the human movement science field there are few publicly available data sets (for example, see *Moore, Hnat & Van den Bogert, 2015* and the references therein). However, none of the available data sets are about human balance. Hence, the purpose of the present study is to describe how a publicly available of balance evaluations on young and elderly adults was created.

## METHODS

This study was designed to create a public repository of data related to human balance, employing quantitative and qualitative evaluations. The entire data collection for each subject was performed in a single session, which lasted between one and two hours. Each subject was assessed by the same experienced examiner (D.A.S.) in the Laboratory of Biomechanics and Motor Control at the Federal University of ABC, Brazil. Prior to the evaluations that generated this data set, we conducted pilot studies with five subjects for training with the equipment and experimental protocol. The data of these subjects are not included in this data set. This study was approved by the local ethics committee of the Federal University of ABC (#842529/2014).

### Subjects

We evaluated a convenience sample of 163 subjects (116 females and 47 males) who voluntarily participated in this study. The subjects were recruited via flyers, advertising on social networks and word to mouth from local communities and included students, professors, and technicians from the university, the local neighborhood, and a community center for older adults. The subjects were first interviewed to collect information about their socio-cultural, demographic, and health characteristics. Their ages varied from 18 to 85 years, body masses from 44.0 to 75.9 kg, heights from 140.0 to 189.8 cm, and body-mass indexes (BMI) from 17.2 to 31.9 kg/m$^2$. Of the 163 subjects, 16 of them were classified as having at least one severe disability or more (eight with hearing and vestibular deficits; two with visual deficits; three with musculoskeletal deficits, one with visual and musculoskeletal

deficits, one with hearing and visual deficits, and one with intellectual disability). All this information for each subject are presented in the public data set (see later on how to obtain it).

## Stabilography

The stabilography evaluation was based on the most common practices used in research laboratories and the clinical environment (for a review see *Duarte & Freitas, 2010*; *Paillard & Noe, 2015*; *Scoppa et al., 2013*; *Visser et al., 2008*).

We evaluated the subjects' balance while standing still for 60 s, in each of four different conditions: on a rigid surface with eyes open; on a rigid surface with eyes closed; on an unstable surface, a 6 cm height foam block (Balance Pad; Airex AG, Sins, Switzerland), with eyes open; on an unstable surface with eyes closed. Each condition was performed three times and the order of the conditions was randomized among subjects. The randomization was performed before the data collection by the examiner using a computerized random number generator. In all conditions, the subjects were required to stand, barefoot and as still as possible with their arms at their sides, and to look at a 5 cm round black target placed on the subject's eye-height on a wall 3 m ahead. For the trials where the eyes were kept closed, subjects were first instructed to look at the target with eyes open, find a stable and comfortable posture given the requirements, and close their eyes. A few seconds later, the data acquisition started. For all trials, the subject's feet were placed with an angle of 20 degrees between them and their heels were kept 10 cm apart by requesting the subjects to stand on lines marked on the top of the force platform (see Fig. 1). The trials were acquired in an empty 4.5 × 2.8 m room with white walls and adequate illumination (see Fig. 1).

### *Protocol*

We followed the following procedure for the stabilography:

1. The researcher explained to the subject about the data collection with the force plate. The subject was also informed that during the data collection, he or she would be monitored, there should not be any verbal communication during the trials, but he or she could interrupt the data collection if desired and that assistance would be given if necessary;
2. The researcher recorded subject's name and identification number, the force platform was zeroed and the subject was requested to stand on the force platform to record her or his weight;
3. The researcher gave instructions on how the subject should stand on the force platform according to the task (open or closed eyes, firm or foam surface). The subject's feet was positioned on the marks at the force platform (see Fig. 1);
4. The researcher instructed the subject to maintain their arms along their body and to stand as still as possible;
5. During the trials with eyes open, the subjects were told to fix their gaze on the round black target placed on the wall ahead at the eye level;
6. During the trials with eyes closed, the subjects were told to fix their gaze at the target placed ahead, close their eyes when they felt ready and only open them when the researcher informed them of end of the trial;

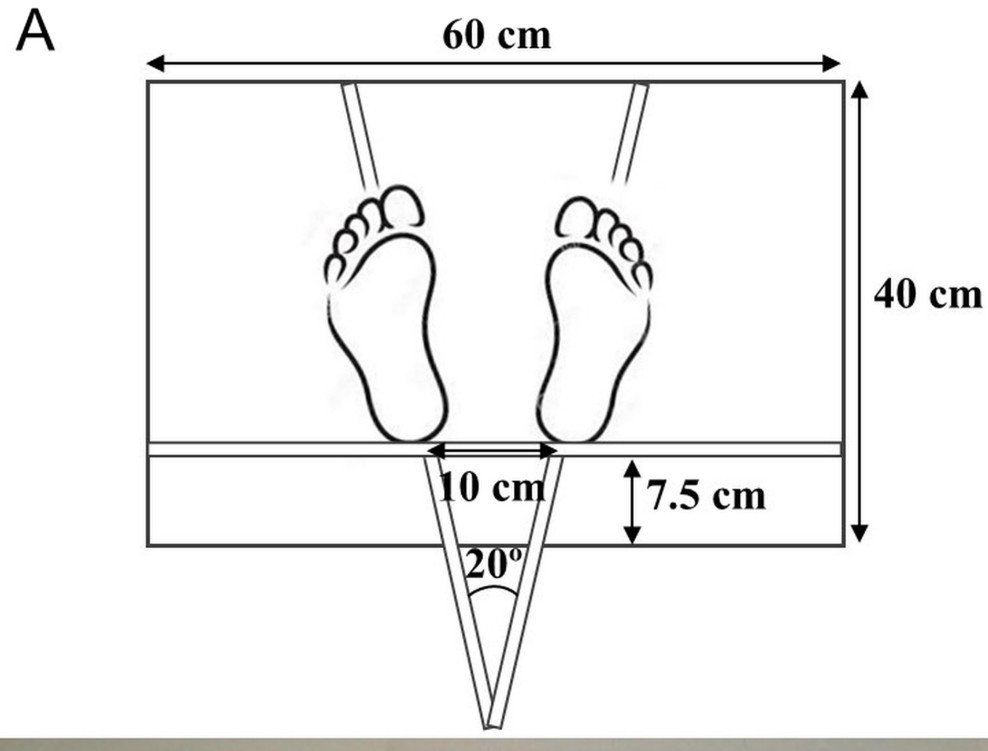

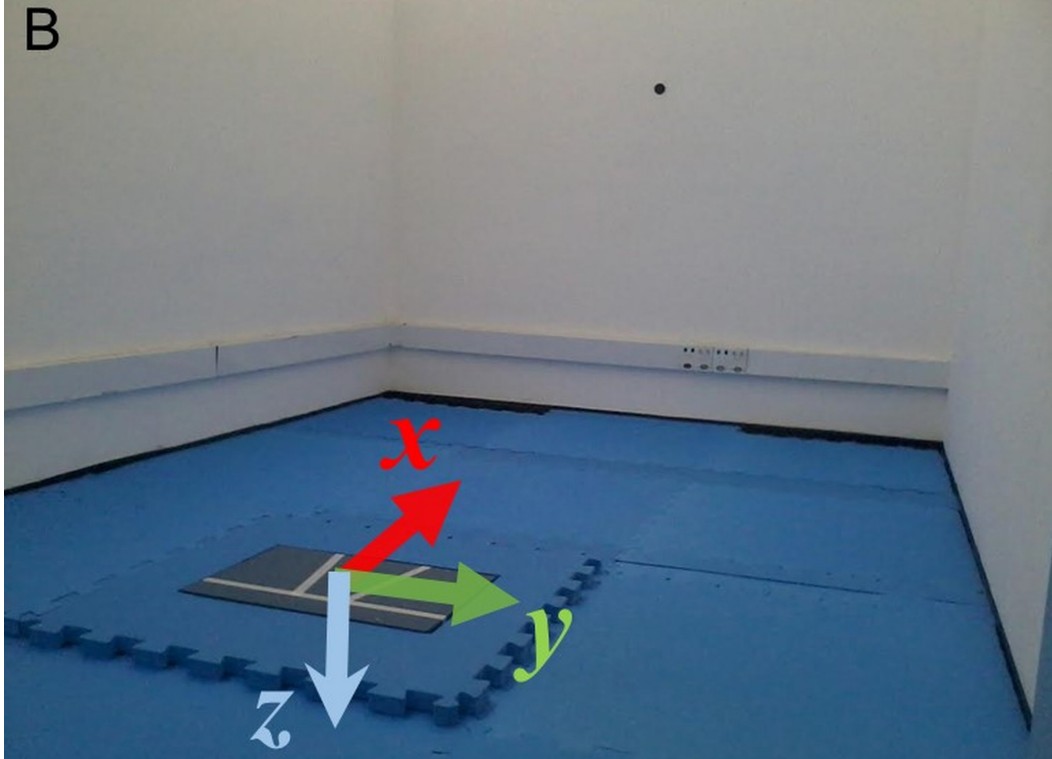

**Figure 1** (A) Marks for the subject's feet placement on the force platform. (B) Data collection room for the stabilography (note the 5 cm black target at the wall 3 m ahead) and the force platform with its coordinate system convention.

7. The researcher started the data collection around 5 s later when the subject said he or she was ready;

8. At the end of the trial, the subject was assisted to step from the force platform and he or she could rest (and sit if desired) for about one minute before the next trial.

9. If the subject was unable to complete a 60 s trial, the test was stopped and that trial was immediately repeated up to two times if necessary. If, after the second attempt, the subject was still unable to complete the 60 s trial, that trial was considered missing.

### Data acquisition and processing

The force platform signals were collected employing a 40 cm × 60 cm commercial force platform (OPT400600-1000; AMTI, Watertown, MA, USA) and amplifier (Optima Signal Conditioner; AMTI, Watertown, MA, USA) at a sampling frequency of 100 Hz. The Optima force plate was factory-calibrated and presented an average COP accuracy of 0.02 cm. However, we did not check this calibration after the force platform installation in our laboratory, and it has been shown that the installation procedure might have deteriorated the force platform calibration (see for example, *Cappello et al., 2011*). The fluctuation of the COP displacement (an indication of the measurement precision) was estimated as the standard-deviation value of the COP data when a 30 kg static load was placed on the force plate for 30 s and was equal to 0.005 cm. The data acquisition was performed employing NetForce software (Version 3.5.3; AMTI, Watertown, MA, USA). The NetForce software outputs the calibrated forces and moments of forces (Fx, Fy, Fz, Mx, My, Mz) from the Optima platform to a file with a proprietary binary format. All subsequent steps, reading these binary files, data processing, analysis and visualization, and exporting data to text files, were implemented in Python language using the SciPy Stack (https://www.scipy.org/) and are available as Jupyter Notebooks (http://jupyter.org/) in a GitHub repository (https://github.com/demotu/datasets).

The force platform data were smoothed with a 10 Hz 4th order zero lag low-pass Butterworth filter and the center of pressure in the anterior–posterior (x-positive is anterior) and medio–lateral (y-positive is to the right) directions (see Fig. 1) were calculated according to the standard formulas (also described in the force platform manual):

$$COP_x = -\frac{M_y}{F_z}$$
$$COP_y = \frac{M_x}{F_z}.$$

Note that the force platform data are expressed as forces and moments of forces in the force platform co-ordinate system and they refer to the forces and moments of forces the subject is applying on the force platform. This is the inverse of the ground reaction forces, where the forces act on the subject (Newton's Third Law).

### Mini balance evaluation systems tests

Each subject's balance was also evaluated by the Mini Balance Evaluation Systems Tests (Mini-BESTest). The Mini-BESTest contains 14 items classified in four different domains of the human balance: Anticipatory Postural Adjustment; Reactive Postural Response;

Sensorial Organization; and Gait Stability; its maximum score is 28 points and each item varies from 0 (abnormal performance) to 2 (normal performance) points (*Franchignoni et al., 2010*). It has been shown that the Mini-BESTest, in general, presents excellent to good reliability and validity (*Potter & Brandfass, 2015*). The tasks in the Mini BESTest requiring the subjects' gaze fixed at a target were performed with the target placed 3 m ahead.

**Other evaluations**

In order to assess the subject's concern about falling, we applied the Short Falls Efficacy Scale International (Short FES-I), which is a scale with seven items. The subjects rank their level of concern that they might fall if doing a hypothetical activity (*Kempen et al., 2008*). The Short FES-I has excellent reliability and validity (*Kempen et al., 2008*). In order to characterize the subject's practice of health-related physical activity, we applied the International Physical Activity Questionnaire—Short Version (IPAQ-SV), which has eight questions about the time the subject spends being physically active (*Craig et al., 2003*). The IPAQ-SV presents acceptable reliability and validity (*Craig et al., 2003*). In order to characterize the subject's cognitive functions, we applied the Trail Making Test (TMT), which instructs the subject to connect a set of dots as quickly as possible (*Reitan, 1958*). In general, the TMT has acceptable reliability and validity (*Poreh et al., 2012*). These evaluations were conducted with personal interviews performed in an empty 7.5 × 5.7 m room with adequate illumination. We followed the instructions of each evaluation test. For the IPAQ-SV, the last seven days prior to the interview were considered. For the TMT, we followed the orientations by (*Bowie & Harvey, 2006*) to administer parts A and B with the following details: the letter "K" was excluded in the evaluation because it is less common to Brazilians and part B of the TMT was composed by 24 circles with numbers from "1" to "12" and letters from "A" to "M."

## RESULTS

All the data is available at PhysioNet (DOI: 10.13026/C2WW2W) and at Figshare (DOI: 10.6084/m9.figshare.3394432). The data at PhysioNet (*Goldberger et al., 2000*) are stored in a binary format that can be read using for example, the WFDB software package or online using the physiobank ATM software and are made available under the ODC Public Domain Dedication and License v1.0 (http://opendatacommons.org/licenses/pddl/1.0/). The data at Figshare are stored in ASCII (text) format that can be downloaded as a single compressed file and are made available under the CC-BY license (https://creativecommons.org/licenses/by/4.0/).

The data set comprises files with the force platform data of 163 subjects plus one file named BDSinfo (in two different formats, .txt or .xlsx) with meta data about each file, subject and the evaluation results. The data files and the BDSinfo.txt file are in ASCII format with tab-separated columns. The files with the force platform data have the following columns: Time, Fx, Fy, Fz, Mx, My, Mz, COPx, COPy, and contain the force platform signals (see Methods). Each file has the following header: Time[s] Fx[N] Fy[N] Fz[N] Mx[Nm] My[Nm] Mz[Nm] COPx[cm] COPy[cm], followed by 6000 rows by 9 columns of data with 6-digit numeric precision (with the exception of the time column,

which has a 3-digit numeric precision). These files are named as BDSxxxxx.txt, where BDS stands for the project's name, Balance Data Set, and xxxxx refers to the number of the trial; from 00001 to 01956. For each subject there are 12 trials and the numbers in the file names are grouped in sets of 12, e.g., the first subject has files from 00001 to 00012 and the last subject (the 163rd), from 01945 to 01956. A total of 26 files (trials) are missing for five subjects who were unable to complete the most challenging conditions.

The BDSinfo file contains meta data describing the conditions of the stabilography trials, the information from the anamnesis, and the results of the qualitative evaluations. Because a subject has 12 files for the force platform data, there are 12 rows for each subject in this file. In these 12 rows, the only column that has rows with different values is the column identifying the trial (the file name). The content of all the other columns are simply repeated over the 12 rows. As result, the BDSdata file has the header plus 1930 rows and 64 columns. Here is the coding for the meta data (the first word identifies the name of the column in the header):

1. **Trial:** file name of the stabilography trial (BDSxxxxx, where xxxxx varies from 00001 to 01956).
2. **Subject:** number of the subject (from 1 to 163).
3. **Vision:** visual condition (Open or Closed).
4. **Surface:** surface support condition (Firm or Foam).
5. **Age:** subject's age in years.
6. **AgeGroup:** age group (Young: **Age** < 60; Old: **Age** $\geq$ 60).
7. **Gender:** gender (F or M).
8. **Height:** height in centimeters (measured with a calibrated stadiometer).
9. **Weight:** weight in kilograms (measured with a calibrated scale).
10. **BMI:** body mass index in kg/m$^2$.
11. **FootLen:** foot length in centimeters (average of the two feet, measured with a calibrated paquimeter).
12. **Nationality:** country where the subject was born.
13. **SkinColor:** self-reported skin color.
14. **Ystudy:** years of regular study.
15. **Footwear:** most common type of footwear the subject wears daily.
16. **Illness:** whether the subject has any illness, as declared by themselves (Yes or No).
17. **Illness2:** type of illness of the subject ('No' if the subjects doesn't have any illness).
18. **Nmedication:** total number of medications the subject takes per day, if any.
19. **Medication:** name of the medication(s) the subject takes ('No' if the subject doesn't take any medication).
20. **Ortho-Prosthesis:** whether the subject wears any type of orthosis or prosthesis, as declared themselves (Yes or No).
21. **Ortho-Prosthesis2:** name of the orthosis or prosthesis the subject wears ('No' if the subject doesn't take any orthosis or prosthesis).
22. **Disability:** whether the subject has any deficit, as declared by themselves (Yes or No).
23. **Disability2:** name of the disability of the subject ('No' if the subject doesn't take any disability).

24. **Falls12m:** how many non-intentional falls the subject had in the last 12 months, as declared by themselves (from 0 to …).
25. **FES_1:** answer for the first question of the Short Falls Efficacy Scale International test (FES-I).
26. **FES_2:** answer for the second question of the FES-I.
27. **FES_3:** answer for the third question of the FES-I.
28. **FES_4:** answer for the fourth question of the FES-I.
29. **FES_5:** answer for the fifth question of the FES-I.
30. **FES_6:** answer for the sixth question of the FES-I.
31. **FES_7:** answer for the seventh question of the FES-I.
32. **FES_T:** answer for the total score question of the FES-I.
33. **FES_S:** answer for the scoring question of the FES-I, see http://www.profane.eu.org/fesi.html.
34. **IPAQ_1a:** answer for the 1a question of the International Physical Activity Questionnaire Short Version test (IPAQ).
35. **IPAQ_1b:** answer for the 1b question of the IPAQ.
36. **IPAQ_2a:** answer for the 2a question of the IPAQ.
37. **IPAQ_2b:** answer for the 2b question of the IPAQ.
38. **IPAQ_3a:** answer for the 3a question of the IPAQ.
39. **IPAQ_3b:** answer for the 3b question of the IPAQ.
40. **IPAQ_4a:** answer for the 4a question of the IPAQ.
41. **IPAQ_4b:** answer for the 4b question of the IPAQ.
42. **IPAQ_S:** score in the IPAQ (Low, Moderate, or High), see https://sites.google.com/site/theipaq/home.
43. **TMT_timeA:** time in seconds taken to complete part A of the Trail Making Test (TMT). We didn't measure times longer than 5 min (for these cases we report a time of 300 s).
44. **TMT_errorsA:** number of errors in part A of the TMT.
45. **TMT_timeB:** time in seconds taken to complete part B of the TMT. We did not measure times longer than 5 min (for these cases we report a time of 300 s).
46. **TMT_errorsB:** number of errors in part B of the TMT.
47. **Best_1:** score for the first task of the Mini Balance Evaluation Systems Test (Mini-BESTest).
48. **Best_2:** score for the second task of the Mini-BESTest.
49. **Best_3l:** score for the third task (left side) of the Mini-BESTest.
50. **Best_3r:** score for the third task (right side) of the Mini-BESTest.
51. **Best_4:** score for the fourth task of the Mini-BESTest.
52. **Best_5:** score for the fifth task of the Mini-BESTest.
53. **Best_6l:** score for the sixth task (left side) of the Mini-BESTest.
54. **Best_6r:** score for the sixth task (right side) of the Mini-BESTest.
55. **Best_7:** score for the seventh task of the Mini-BESTest.
56. **Best_8:** score for the eighth task of the Mini-BESTest.
57. **Best_9:** score for the ninth task of the Mini-BESTest.
58. **Best_10:** score for the tenth task of the Mini-BESTest.

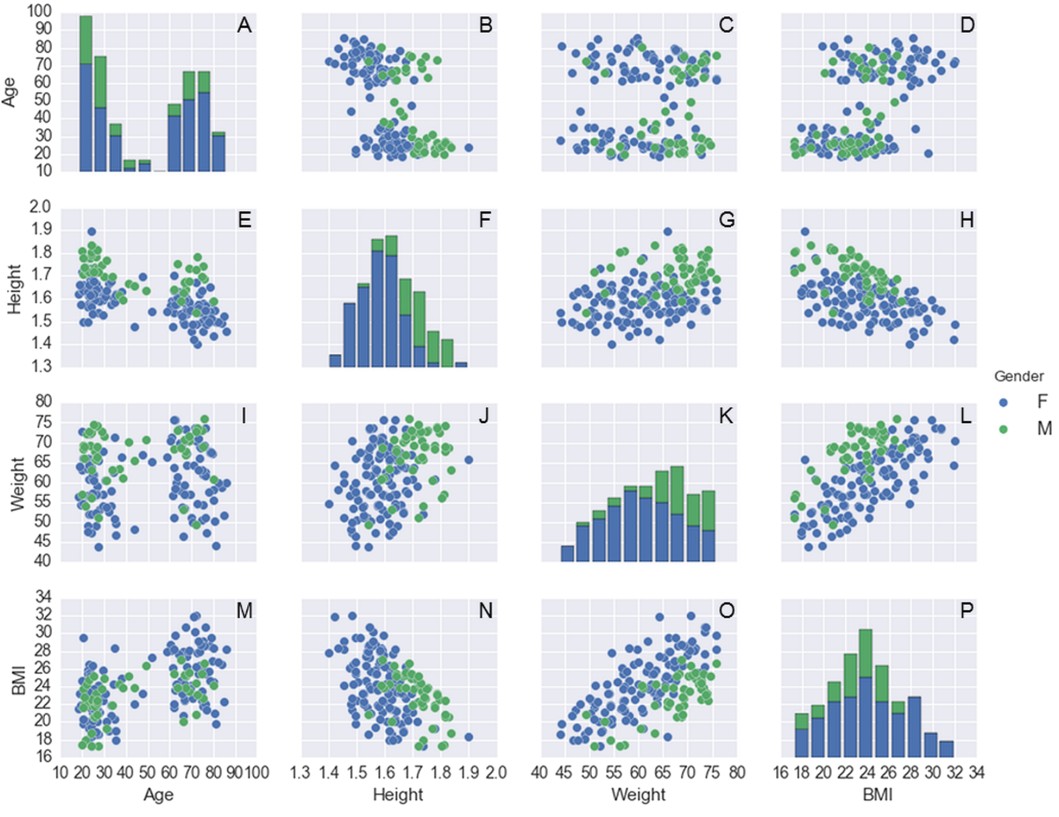

**Figure 2** **Histogram plots on the main diagonal and scatter plots outside the diagonal for the following variables: age (years), body mass (kg), height (m), and BMI (kg/m²) for each subject.** These are color-coded by gender (116 females and 47 males). Note that, for the histograms on the main diagonal, the actual vertical axes (the frequencies of observations occurring in certain ranges of the variables shown on the bottom horizontal axes) are not shown. The name of the variables plotted in each scatter plot outside the main diagonal are shown in the legends on the left side and the bottom of the 4 × 4 panel.

59. **Best_11:** score for the eleventh task of the Mini-BESTest.
60. **Best_12:** score for the twelfth task of the Mini-BESTest.
61. **Best_13:** score for the thirteenth task of the Mini-BESTest.
62. **Best_14:** score for the fourteenth task of the Mini-BESTest.
63. **Best_T:** total score of the Mini-BESTest, see http://www.bestest.us/.
64. **Date:** date and time of the subject's evaluation (yyyy-mm-dd hh:mm:ss.sss 24-hour local time format).

## Data exploration

To exemplify how these data can be explored, shown in Fig. 2 are histograms and scatter plots for the variables: age, body mass, height, and BMI, and in Fig. 3 are similar plots for the variables: FES_T, Best_T, IPAQ_S, TMT_timeA, and TMT_timeB. A representative example of the force platform data is shown in Fig. 4 and plots for the variables: COP area, COP velocity, and COP mean frequency for each subject and standing condition are shown in Fig. 5 (see *Duarte, 2015*; *Duarte & Freitas, 2010* for the algorithms to calculate these variables). The programming scripts to generate these figures, as well

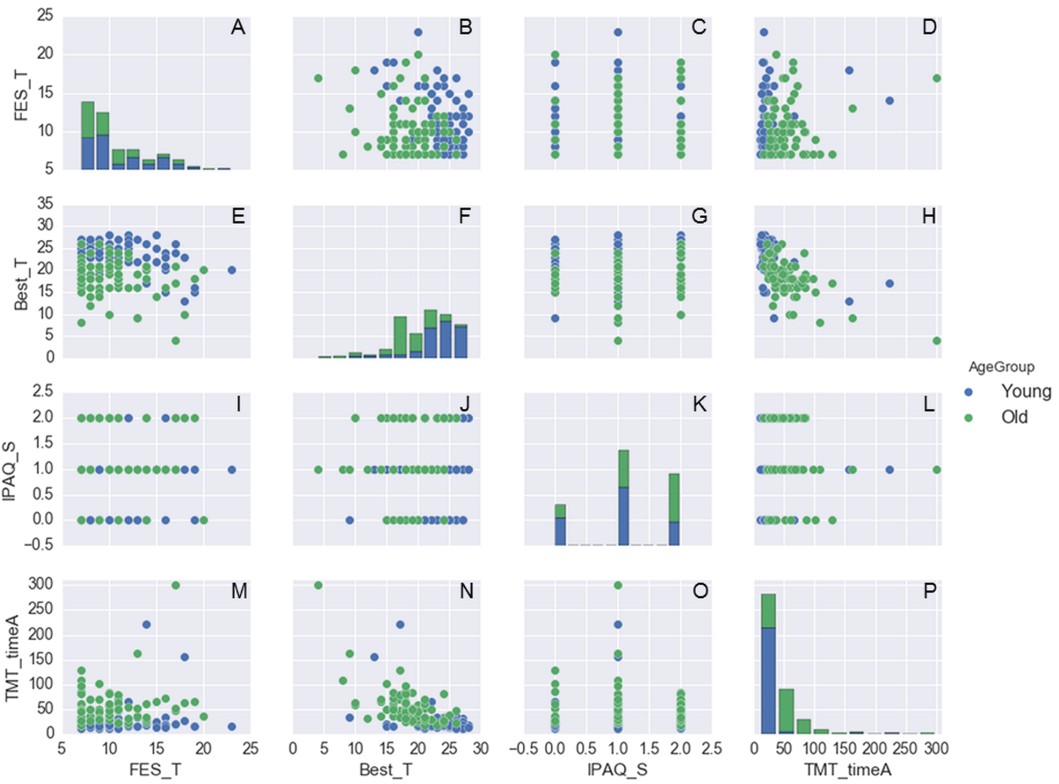

**Figure 3** **Histogram plots on the main diagonal and scatter plots outside the diagonal for the following variables: FES_T, Best_T, IPAQ_S, and TMT_timeA. These are color-coded by age group (87 young adults and 76 older adults).** Note that, for the histograms on the main diagonal, the actual vertical axes (the frequencies of observations occurring in certain ranges of the variables shown on the bottom horizontal axes) are not shown. The name of the variables plotted in each scatter plot outside the main diagonal are shown in the legends on the left side and the bottom of the 4 × 4 panel.

as other examples on how to work with these data, are available in a GitHub repository (https://github.com/demotu/datasets).

## DISCUSSION

This study provides a publicly available data set with qualitative and quantitative evaluations related to the balance of young and elderly adults (a total of 163 subjects). All the data is available at PhysioNet (DOI: 10.13026/C2WW2W) and at Figshare (DOI: 10.6084/m9.figshare.3394432).

A limitation of this study is that we did not perform quantitative evaluations of the subjects' health conditions, particularly of their motor and sensory integrity. However, through the careful anamnesis we performed by interview, we were able to determine the history and current status of the subjects' health, and together with the different qualitative evaluations performed, we believe the subjects are relatively well characterized. Nevertheless, the potential user of this data set should bear this limitation in mind. We studied a convenience sample of subjects who voluntarily accepted to participate in our

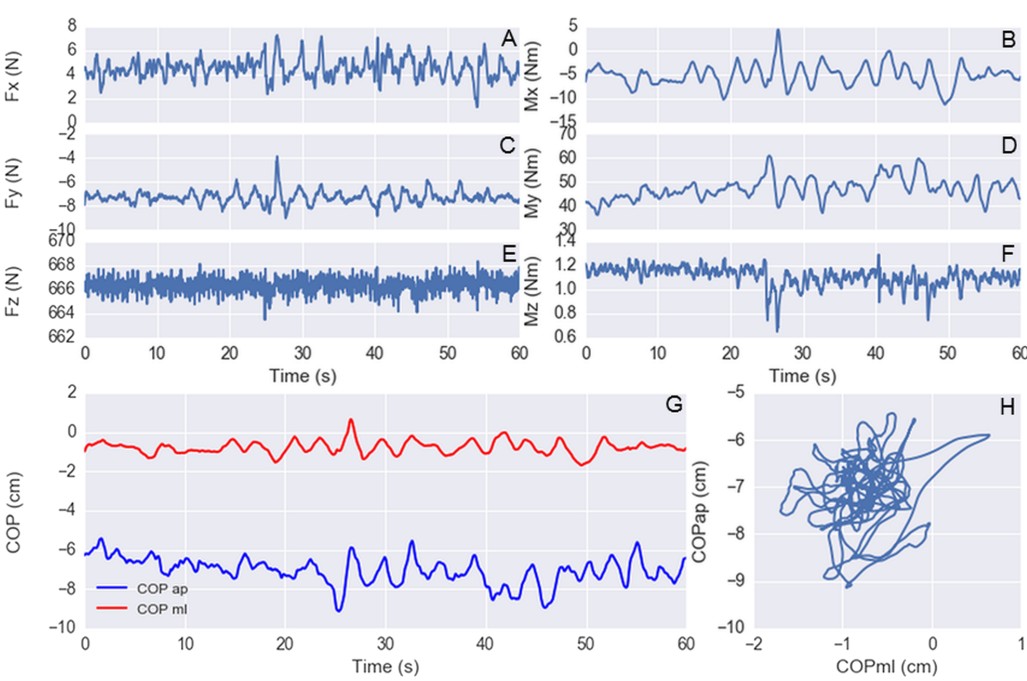

**Figure 4** (A–F) exemplary forces and moments of forces at the horizontal and vertical anterior-posterior (x), medio-lateral (y), and vertical (z) directions versus time. (G) center of pressure (COP) displacements at the ap and ml directions versus time. (H) plot of COP ap versus COP ml. Subject: 61-year-old male adult during quiet standing on a rigid surface with open eyes.

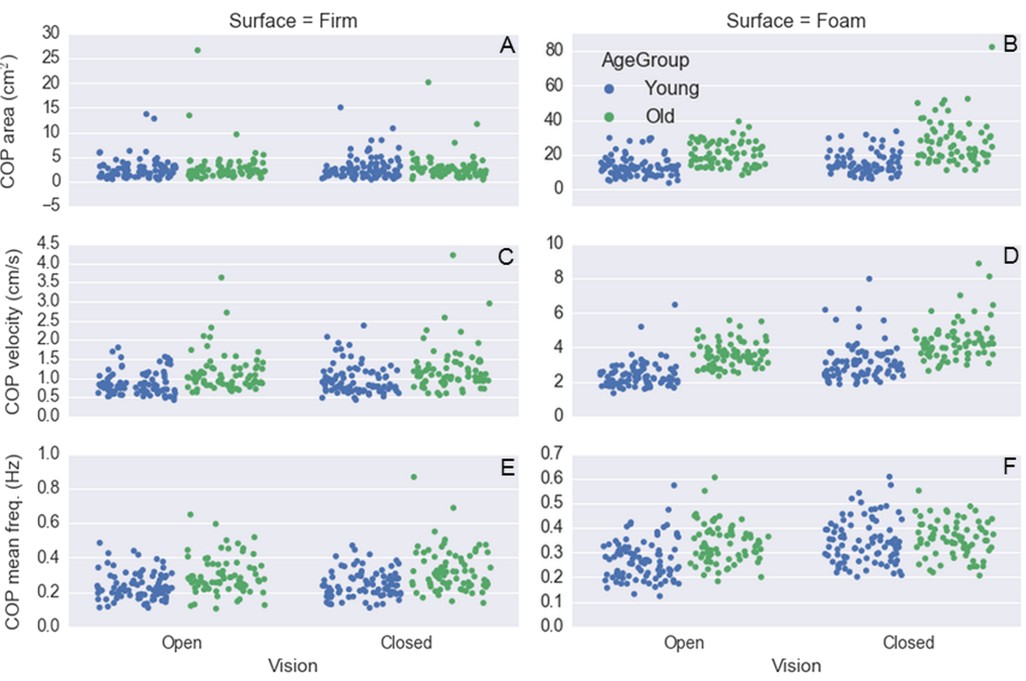

**Figure 5** Plot of the median value across trials of the variables: COP area, resultant COP velocity, and resultant COP mean frequency for each subject at the different visual and support surface conditions color-coded by age group (87 young adults and 76 older adults).

study, which was conducted at a university laboratory; as such, the sample might be unintentionally biased. Even though, we classified 16 of the 163 investigated subjects as people with disabilities. If one is looking for reference data on balance by healthy people, the data of these 16 subjects should be excluded. One unintended characteristic of the data set is that the subjects can be roughly grouped in two age groups: 20–40 and 60–90 years old.

The Balance Data Set is the first public repository containing data of quantitative and qualitative evaluations of human balance. Possible applications of this data set include: to test new variables to describe the center of pressure displacement in methodological studies; to serve as reference (normative) data for a new sample of subjects in a research of clinics context; for training and education regarding the analysis of balance data, among others. Examples on how to work with this data set are publicly available in a GitHub repository (https://github.com/demotu/datasets).

### Funding
This study was supported by Fundação de Amparo à Pesquisa do Estado de São Paulo from Brazil (#13/26829-1 and #14/13247-7) and Conselho Nacional de Desenvolvimento Científico e Tecnológico from Brazil (484464/2013-2). The funders had no role in study design, data collection and analysis, decision to publish, or preparation of the manuscript.

### Grant Disclosures
The following grant information was disclosed by the authors:
Fundação de Amparo à Pesquisa do Estado de São Paulo: #13/26829-1, #14/13247-7.
Conselho Nacional de Desenvolvimento Científico e Tecnológico: 484464/2013-2.

### Competing Interests
The authors declare there are no competing interests.

### Author Contributions
- Damiana A. Santos performed the experiments, analyzed the data, wrote the paper, prepared figures and/or tables, reviewed drafts of the paper.
- Marcos Duarte conceived and designed the experiments, analyzed the data, wrote the paper, prepared figures and/or tables, reviewed drafts of the paper.

### Human Ethics
The following information was supplied relating to ethical approvals (i.e., approving body and any reference numbers):
Ethics Committee of the Federal University of ABC (#842529/2014).

### Data Availability
Human Balance Evaluation Database:
DOI: 10.13026/C2WW2W;

Santos, Damiana A. dos; Duarte, Marcos (2016): A public data set of human balance evaluations. Figshare. https://dx.doi.org/10.6084/m9.figshare.3394432.v2.

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
