# Peer review of "A public data set of human balance evaluations"

_PeerJ, doi:10.7717/peerj.2648_

## Round 0.1 · original submission · Minor Revisions

· Academic Editor

Minor Revisions

Both reviewers were quite positive about your work overall. They both have provided similar feedback especially about the sample characteristics. Please carefully address all their concerns in your revised version.

·

Basic reporting

No comments

Experimental design

No comments

Validity of the findings

No comments

Additional comments

The manuscript is clear and well written and I believe the work presented could be of great value to the posturography community. I do however believe there are some issues that should be addressed.

The biggest limitations I find in the work is the fact that the authors decided, with no good reason I can think of, to publish in the dataset only the force platform data filtered at 10 Hz. As they rightly mention in the introduction, “In stabilography, there is no consensus yet on the best techniques to analyze COP displacement in order to extract meaningful information about the subject’s balance condition.” How to filter the data is one of the issues where no consensus has jet been reached. For instance, in a preliminary investigation (Oggero E, Carrick FR, Pagnacco G. Frequency content of standard posturographic measures. Biomed Sci Instrum. 2013;49:48-53.) frequency content of Fz and CoP was found for some subjects to be over 20 Hz. It would be much more useful if the data published in the public dataset were not filtered, leaving investigators using them to experiment with different algorithms on the raw data. In general, I believe that public datasets like the one the authors envision to aid in the development of protocols and techniques should contain data processed as little as possible so not to risk loosing information that could be important.

Other issues:

It is unclear what were the recruitment criteria and modalities. Did the authors seek healthy volunteers, pathological subjects, or just anyone willing to participate? Where the subjects recruited from the local community or referred from clinicians (therefore possibly with some health complaint, even if apparently unrelated to postural balance)? Although the dataset contains information that can be used to answer some of these questions, it should be clearly specified in the manuscript.

Line 133: “The Optima force plate was factory-calibrated and presented an average COP accuracy of 0.02 cm.” Unless the authors actually verified the accuracy, this is incorrectly stated and possibly misleading: the force platform was factory calibrated at that level of accuracy, but it is likely that that is not the level of accuracy once the platform is installed (because of the stresses and deformations caused by the installation). If the platform has been in use for a time, its calibration bight also have been affected by aging and stresses, possibly reducing the accuracy. See, for example, the following references:
Cappello A, Bagalà F, Cedraro A, Chiari L. Non-linear re-calibration of force platforms. Gait Posture. 2011 Apr;33(4):724-6. doi: 10.1016/j.gaitpost.2011.02.008. Epub 2011 Mar 9.
Chockalingam N, Giakas G, Iossifidou A. Do strain gauge force platforms need in situ correction? Gait Posture 2002;16:233–7.
Gill HS, O’Connor JJ. A new testing rig for force platform calibration and accuracy tests. Gait Posture 1997;5:228–32.
Hall MG, Fleming HE, Dolan MJ, Millbank SF, Paul JP. Static in situ calibration of force plates. J Biomech 1996;29(5):659–65.
I recommend rephrasing the sentence, or if possible, to actually determining the average accuracy.

Throughout the document: the correct unit abbreviation for kilograms is kg, not Kg. Also, there should be no "–" between the number and the units abbreviation, e.g., line 136 “30 kg” not 30-kg”, and in line 144, “10 Hz” rather then “10-Hz”.

Reference to commercial materials/devices should contain the manufacturer’s name, city/town, state (if applicable) and country, e.g., “(OPT400600-1000, AMTI, Watertown, MA, USA)”, as there could be another manufacturer of the same name in other locations but providing different materials/devices (for instance there are many US companies under the AMTI abbreviation, albeit in different fields)

Line 131: the dimensions of the force platform are not part of the make/model, therefore they should be outside the parenthesis referencing the device. Suggest rephrasing as “The force platform signals were collected at a sampling frequency of 100 Hz by employing a 40 cm x 60 cm commercial force platform (OPT400600-1000, AMTI, Watertown, MA, USA) and amplifier (Optima Signal Conditioner, AMTI, Watertown, MA, USA)…”

References: most references have a DOI, but it is unclear. As suggested in the Instructions For Authors, I recommend adding “DOI:” in front of the actual DOI reference. Example journal reference: Smith JL, Jones P, Wang X. 2004. Investigating ecological destruction in the Amazon. Journal of the Amazon Rainforest 112:368-374. DOI: 10.1234/amazon.15886.

Reviewer 2 ·

Basic reporting

This paper could do with some extra editing to make it easier to read and improve clarity.
Figures 2 and 3, used to illustrate the data, are quite confusing, as the authors have tried to show too much data.

Experimental design

The aim needs to be made clearer that the paper sets out to describe the process of how the data was created.
The authors have not mentioned that the age profile of the subjects is very much 20 -30 and 60 plus with very little in the 40 -60 age group. Furthermore they divided the set into Young (under 60) and Old (over 60) without explaining why this cut-off is used.
Perhaps there is data to explain why this cut-off is used?
How were the subjects recruited? If they were all connected to the place of study (clinic or university) it may reflect a bias selection process?
There were 16 people with reported health problems. Are these the outliers seen is some of the graphs? Would the data set be better if these were removed and then it could reflect a healthy data set for comparison against others?
When and how was the randomization of tests done and by whom?
Also what happened to the people who felt unsteady or stepped off the mark doing the test (especially the eyes closed)? It is highly unlikely that someone didn’t wobble? Did they start again or stop the test at that point?

Can the authors explain why they selected the: Short Falls Efficacy Scale International, International Physical Activity Questionnaire - Short Version and Trail Making Test (TMT) for the qualitative test and provide some details on the reliability or validity of the tests?

Validity of the findings

As all the data is already publicly available, it is not relevant to comment on this.
Figure 4 is very useful to show how the data can be interpreted.
Some basic tests of normality might be more useful to illustrate the normal distribution of the data that would confirm its validity.
While the histograms are illustrative, some of the tests should be skewed, as the subjects are quite young so would be expected to score better on tests that are designed to detect pathology.

Additional comments

I welcome this attempt to create a publicly available data set for researchers to access and hope this will continue.

Annotated reviews are not available for download in order to protect the identity of reviewers who chose to remain anonymous.

---

## Round 0.2 · accepted · Accept

· Academic Editor

Accept

The reviewers were satisfied with your responses.

·

Basic reporting

No Comments

Experimental design

No Comments

Validity of the findings

No Comments

Additional comments

I thank the authors for addressing all the issues my fellow reviewer and I indicated in the previous revision.